# The Pharmacological Treatment of Chronic Pain: From Guidelines to Daily Clinical Practice

**DOI:** 10.3390/pharmaceutics15041165

**Published:** 2023-04-06

**Authors:** Gianmarco Marcianò, Cristina Vocca, Maurizio Evangelista, Caterina Palleria, Lucia Muraca, Cecilia Galati, Francesco Monea, Liberata Sportiello, Giovambattista De Sarro, Annalisa Capuano, Luca Gallelli

**Affiliations:** 1Operative Unit of Pharmacology and Pharmacovigilance, “Mater Domini” Hospital, 88100 Catanzaro, Italy; 2Department of Anesthesia, Resuscitation and Pain Therapy, Sacred Heart Catholic University, 00100 Rome, Italy; 3Department of Primary Care, ASP 7, 88100 Catanzaro, Italy; 4Research Center FAS@UMG, Department of Health Science, University Magna Graecia, 88100 Catanzaro, Italy; 5Campania Regional Centre for Pharmacovigilance and Pharmacoepidemiology, 80138 Naples, Italy; 6Department of Experimental Medicine, Section of Pharmacology “L. Donatelli”, University of Campania “Luigi Vanvitelli”, Via Costantinopoli 16, 80138 Naples, Italy; 7Medifarmagen Srl, University of Catanzaro and Mater Domini Hospital, 88100 Catanzaro, Italy

**Keywords:** nociceptive pain, neuropathic pain, nociplastic pain, drug treatment, drug interactions, guidelines

## Abstract

In agreement with the International Association for the Study of Pain, chronic pain is an unpleasant sensory and emotional experience associated with actual or potential tissue damage. To date, there are several types of pain: nociceptive, neuropathic, and nociplastic. In the present narrative review, we evaluated the characteristics of the drugs used for each type of pain, according to guidelines, and their effects in people with comorbidity to reduce the development of severe adverse events.

## 1. Introduction

The International Association for the Study of Pain (IASP) describes chronic pain as an unpleasant sensory and emotional experience associated with, or resembling that associated with, actual or potential tissue damage [1]. Chronic pain lasting for at least 3 months may impair the social, psychological, and physical sphere of a subject, leading to serious impairment of both their autonomy and mood [2]. Social factors (e.g., job, lifestyle, and economic and religious status), self-perception, mood alterations, and physical illness are risk factors for chronic pain [3]. According to IASP recommendations, chronic pain is classified as nociceptive (involving tissue or potential tissue damage), neuropathic (involving disease or injury affecting the nervous system), and nociplastic (with no evidence of tissue or nerve damage but persistent overregulation of the nociceptive system) [3,4]. 

Concerning the types (mechanisms) of pain, the choices of drugs for chronic pain include non-steroidal anti-inflammatory drugs (NSAIDs) (for cyclic short-course treatment), opioids, and central nervous system (CNS)-acting drugs. 

### 1.1. NSAIDs

In patients with chronic nociceptive pain (e.g. tendonitis, osteoarthritis, or back pain), a shorter course of NSAIDs can be considered [5,6,7]. Evaluating their pharmacodynamic differences, ibuprofen and naproxen are non-selective COX inhibitors, and celecoxib and diclofenac are COX-2 semi-selective drugs, whereas etoricoxib is a COX-2 selective inhibitor [8,9]. 

Acetaminophen (paracetamol), an atypical NSAID without anti-inflammatory effects, was reported to be inefficient in patients with persistent chronic low back pain [10].

### 1.2. Opioids

Opioids can be used in patients with nociceptive or neuropathic pain (Table 1) but not in patients with nociplastic pain [11,12] (Figure 1).

### 1.3. Central-Nervous-System-Acting Drugs 

In patients with neuropathic pain, both antidepressants and antiepileptic drugs represent the first-line treatment [13].

Among tricyclic antidepressants, amitriptyline (25–125 mg/day) shows the better number needed to treat for 50% patient relief (with an NNT of 3.6), while duloxetine (30–60 mg/day) has an NNT of 6.4 [14].

Venlafaxine is not commonly used because it has low activity on the noradrenaline pathway with a dosage of lower than 150 mg per day [15]. The mechanism of action of antidepressants is summarized in Figure 2.

Gabapentin and pregabalin act on the α-2-delta subunit of voltage-gated calcium channels. The result of this action is the reduction in substance P, glutamate, and noradrenalin release [16]. Dizziness and somnolence are common adverse events that may lead to dropout. Both these drugs are excreted by the kidneys, and their dosage should be titrated carefully according to the estimated glomerular filtration rate (eGFR) [16,17].

Gabapentin has important disadvantages in comparison with pregabalin since it shows more adverse effects (leukopenia, hypertension, vasodilation, and skin reactions) and has a complex posology/titration scheme, reducing compliance [16,18,19,20].

Among antiepileptics, carbamazepine can be used for the management of chronic neuropathic pain [12], even if, usually, it is commonly prescribed for trigeminal neuralgia [21]. Carbamazepine is associated with several side effects: hematopoietic alterations, hyponatremia, neurologic effects (dizziness, somnolence, headache, and diplopia), gastrointestinal symptoms, cutaneous side effects, fatigue, and an increase in hepatic enzymes. Moreover, it is associated with teratogenic risk in pregnancy. Furthermore, it induces CYP3A4 and auto-induces its own metabolism, which is mediated by the same isoform [22,23]. 

Antiepileptic drugs (i.e., gabapentin and pregabalin) are indicated for neuropathic pain including postherpetic neuralgia, diabetic peripheral neuropathy, and spinal cord injury (with an NNT range of 2.9–7.7) [14]. A topical patch of 5% lidocaine (NNT = 4.4) and 9% capsaicin (NNT = 10.6) can reduce both allodynia and spontaneous pain [24,25]. 

For the management of nociplastic pain, commonly used drugs with neuropathic activity are, e.g., pregabalin and duloxetine [26]. The mechanisms of action of other drugs used for neuropathic pain treatment are summarized in Figure 3. 

### 1.4. Muscle Relaxants 

Muscle relaxants (e.g., cyclobenzaprine, tizanidine, thiocolchicoside, baclofen, eperisone, metaxalone, and methocarbamol) inhibit γ motoneurons and increase the activity of calcium channels and calmodulin with an increase in blood flow in the areas with muscle contraction. Moreover, each muscle relaxant has a particular mechanism of action: eperisone is a substance P antagonist; cyclobenzaprine operates via α and γ motoneurons inhibition; tizanidine is an α_2_ agonist; baclofen is an agonist of γ-aminobutyric acid (GABA)-B receptors; and thiocolchicoside is an agonist of GABA and glycine receptors.

In clinical trials, these drugs show good clinical efficacy, although adverse drug reactions (ADRs) and drug interactions (DDIs) can limit their use [27,28,29] (Table 2), reducing adherence to the drug therapy. 

Even if several guidelines describe the effects of these drugs in patients with chronic pain [2,12,82,83,84], few data related to the treatment of patients with chronic pain and in polytherapy have been published. The aim of the present narrative review is to evaluate the characteristics of the drugs used for each type of chronic pain, according to the guidelines, and their effects in people with comorbidity to reduce the development of severe adverse events.

## 2. Materials and Methods

The PubMed, Embase, and Cochrane library databases were searched for articles published until 10 January 2023, in agreement with our recent papers [55,57,85,86,87,88]. The secondary search included articles cited in the reference lists of papers identified with the primary search. The records were first screened by title/abstract (G.M., C.V., and C.P.), and then full-text articles were retrieved for eligibility evaluation (M.E.). The remaining articles were then subject to a citation search of all reference lists (A.C., L.G., and G.D.S.). Papers were deemed eligible if they included any of the words “chronic non cancer pain”, “drug(s)”, “guidelines”, and “adverse drug reaction”. All citations were downloaded into Mendeley, and duplicates were deleted. To avoid a bias of exclusion, the full-text articles were retrieved following the first round of exclusions and were also subject to two independent eligibility reviews, this time in perfect agreement. The studies evaluated as eligible were included in the present review. We excluded manuscripts without full texts and without indications of effects on chronic pain and manuscripts not in the English language. 

## 3. International Guidelines

The first guidelines for pain treatment are presented in the World Health Organization (WHO) guidelines that, published in 1986, do not separate between both types (acute or chronic) and the different mechanisms (nociceptive, neuropathic, or nociplastic) of pain and suggest a 3-step treatment: NSAIDs (acetaminophen and other NSAIDs; Step I), weak opioids (5 mg of codeine, tramadol, and oxycodone; Step II), or strong opioids (Step III) [89]. 

Other international guidelines separate the types of pain, and in the presence of chronic pain, suggest a multimodal step-by-step approach, also considering the mechanisms of the pain. 

The Centers for Disease Control and Prevention (CDC) guidelines report that chronic pain should primarily be managed with non-opioid drugs [90]. When utilized for pain management, opioids should be started at the lowest effective dosage and titrated slowly [90] (Table 3). 

### 3.1. Neuropathic and Nociceptive Chronic Pain Treatment 

The Scottish Intercollegiate Guidelines Network (SIGN) guidelines [12] suggest a seven-step treatment (Table 3) from history (Step I) to drug treatment (Step IV). To improve drug safety, the authors invite us to evaluate the mechanism of the pain (nociceptive and/or neuropathic pain) and the comorbidity. Step VII suggests an accurate follow-up for exacerbation management [12]. 

The American Pain Society suggests a multi-modal approach without a differentiation concerning the mechanisms of pain [95]. The authors suggest that opioids (mainly taken via the oral route and with caution in opioid-naïve patients), gabapentin and pregabalin, NSAIDs, and paracetamol are possible options [95].

The Colorado Division of Workers Compensation guidelines [49] suggest a drug reconciliation to avoid interaction or prescription errors. In patients with nociceptive pain, the authors suggest a cyclic treatment with NSAIDs for up to 7 days with non-selective NSAIDs and up to 10 days with COX-2 inhibitors. In patients with neuropathic pain, the authors suggest a four-step process (Table 3), supporting the combination of two drugs from different categories to reduce dosage and side effects (e.g., duloxetine plus pregabalin).

Concerning opioids, no evidence of the superiority of one opioid compared with other drugs has been reported. Long-acting opioids have been found to not be superior to short-acting opioids. Among these compounds, oxycodone seems to be the most abused drug.

Buprenorphine has similar efficacy in comparison with tramadol in patients with moderate–severe musculoskeletal pain and with fentanyl (regarding analgesia and sleep quality) for severe pain. 

Muscle relaxants are not suggested for patients with chronic pain due to the habit-forming risk, respiratory depression, and seizure occurrence after sudden withdrawal. 

Topical agents including 8% capsaicin (for postherpetic neuralgia), 5% lidocaine plasters, or 8% pump sprays (for diabetic neuropathy and post-herpetic neuralgia), and 0.1% clonidine (for diabetic peripheral neuropathy) can also be used. Among the new compounds, alpha-lipoic acid (600 mg/die for 3–5 weeks) may be used to manage neuropathic pain.

Trigger point injections (of local anesthetics with or without corticosteroids or needling alone) are a possible option for myofascial pain. 

In elderly patients, the management of pain is detailed in the American Geriatric Society guidelines [91]. For chronic nociceptive pain, paracetamol (up to 4 g daily) should be the first-line medication. NSAIDs can be used in patients that have experienced failure of efficacy or the development of side effects during paracetamol treatment. Opioids can be used in patients with moderate–severe pain and functional impairment, unresponsiveness to NSAIDs, or contraindications to their use (gastritis, severe liver or renal diseases, or allergy to NSAIDs). Patients should also be assessed for the presence of drug toxicity and drug–drug interaction risks [91]. 

In patients with neuropathic pain or fibromyalgia, duloxetine pregabalin or gabapentin can be used even if they must be evaluated for the development of side effects; in contrast, tricyclic antidepressants should be avoided due to their high potential for side effects. Combination therapy seems to increase efficacy and reduce toxicity [91]. 

More recently, the Department of Health and Human Services’ best practices give further information [92], recommending non-opioid or non-pharmacologic therapeutic options in order to avoid chronic treatment with these compounds. For neuropathic pain, the first-line therapy should be chosen among anticonvulsants, SNRIs, amitriptyline, and topical analgesics (capsaicin and lidocaine). For non-neuropathic, non-cancer pain, NSAIDs and paracetamol are the first-line options. Based on patients’ responses, other medication classes include muscle relaxants [92]. Trigger point injection (dry needling injection of local anesthesia) may be useful for the management of headache-associated pain, myofascial pain, and low back pain. 

### 3.2. Neuropathic Chronic Pain

The NeuPSIG guidelines’ last recommendations were published in 2015 [14]. A literature revision was conducted (of 229 studies), performing a meta-analysis that evaluated the number needed to treat (NNT) for 50% patient pain relief. The trial outcomes were poor or modest even for first-line drugs, which was possibly due to overestimations of the placebo effect, scarce patient profiling, and inadequate diagnostic criteria. The new recommendations are summarized in Table 3. The data for the other drugs including other antiepileptics, antidepressants, cannabinoids, tapentadol, and other topical drugs were considered inconclusive. Drugs such as levetiracetam and mexiletine are contraindicated.

The NICE guidelines provide recommendations on chronic neuropathic primary pain (including fibromyalgia). They suggest the use of antidepressants in people ≥ 18 years after the careful evaluation of risk–benefit. Pregabalin or gabapentin and local anesthetics are not suggested, except for in trials for complex regional syndrome. The contraindicated drugs are benzodiazepines, antiepileptics, corticosteroids, trigger point injections, ketamine, NSAIDs, opioids, and paracetamol [2].

The PRACTICE guidelines [94] suggest the use of anticonvulsants, SNRIs, or TCAs in patients with neuropathic pain, with low evidence for the use of SSRIs, NMDA receptor antagonists (e.g., memantine or dextromethorphan), opioids, and muscle relaxants. Topical agents such as capsaicin, lidocaine, and ketamine are also possible options for neuropathic pain. Concerning trigger point injection, it may be considered a multi-modal approach option in patients with myofascial pain (Table 3). 

### 3.3. Nociceptive Chronic Pain 

In patients with knee osteoarthrosis (nociceptive pain), the *ESCEO* group suggests a six-step treatment: chondroitin sulfate and glucosamine sulfate (first step) with or without topical NSAIDs or paracetamol (second step), oral NSAIDs (third step), intra-articular injection of hyaluronic acid or corticosteroids (fourth step), oral SNRI (fifth step), and, finally, surgery (final step) [93].

### 3.4. Nociplastic Pain

To date, there are no definitive guidelines for nociplastic pain; however, considering its pathogenetic mechanism, antidepressants and pregabalin or gabapentin can be used. NSAIDs can be indicated only in the presence of clinical evidence of inflammation, while opioids are not indicated. In fact, in these patients, it has been suggested that higher concentrations of endogenous opiates and opioid use can worsen hyperalgesia and modify sleep architecture [96,97]. For nociplastic pain (e.g., fibromyalgia, chronic back pain, and complex regional pain) a low dose of naltrexone, an opioid antagonist, by increasing the density of opioid receptors, improved the response to endogenous opiates with an improvement in clinical symptoms [98]. A similar activity could be obtained using methadone, a potent MOPR agonist and weak NMDA receptor antagonist, which seems able to reduce opioid-induced hyperalgesia [99].

## 4. From Guidelines to Daily Use: The Problem of Safety in Patients with Comorbidity

Since the guidelines suggest better treatments in the presence of a patient with chronic pain, it is possible that this new drug increases the risk of side effects or drug interactions [49] (Table 2).

A detailed clinical evaluation and history allow the formulation of a proper diagnosis and the description of the pain type [3]. Clinicians should be aware of contraindications for each drug and patient comorbidities since this information is essential in therapeutic decision making. 

### 4.1. Kidney Diseases

Some drugs (e.g., gabapentin and pregabalin) need dose adjustments, whereas other molecules must be avoided, especially for long periods (NSAIDs). Amitriptyline is largely excreted by the kidneys, but its serum levels do not seem to change in chronic kidney disease (CKD), similar to carbamazepine [31,100]. Few data are available for the usage of SNRIs in CKD. Duloxetine metabolites are mainly eliminated in the urine [41,100]. A review by Davison et al. suggests that codeine, morphine, oxycodone, tramadol, and hydrocodone should be avoided in advanced CKD (Table 4). Buprenorphine, hydromorphone, fentanyl, and methadone are the suggested compounds due to their inactive metabolism and scarce kidney excretion. Moreover, none of them are significantly removed using dialysis [101]. 

### 4.2. Hepatic Failure 

Paracetamol is a well-known hepatotoxic drug since all NSAIDs can induce liver injury [63,114]. Dastis et al. [114] suggested a reduction in the paracetamol dose to a maximum of 2 g/daily in patients with non-alcoholic cirrhotic liver disease, avoiding its coadministration with alcohol (due to the increased production of N-acetyl-p-benzoquinone imine (NAPQI)). However, in our opinion, this is a risk, and caution is recommended in patients with hepatic insufficiency, hepatitis, concomitant treatment altering hepatic function, a deficit in glucosium-6-phospate-dehydrogenase (GSPD), and hemolytic anemia [63]. The use of NSAIDs in cirrhosis or hepatic impairment may be very dangerous. Aspirin shows dose-dependent liver toxicity, and other drugs (nimesulide, diclofenac, and sulindac) are also associated with this effect. Nevertheless, hepatotoxicity is a class effect, which is often related to idiosyncratic mechanisms. Furthermore, NSAIDs may increase the bleeding risk and the worsening of kidney function in hepatorenal syndrome. Their interaction with diuretics used in the management of cirrhosis may further impair renal effectiveness and act as an obstacle to the management of ascites. Aspirin is contraindicated in cases of severe hepatic insufficiency. Caution and, eventually, dose adjustment are required for other NSAIDs [114,115]. 

Opioids are also metabolized by liver cytochromes, and caution is required in clinical practice because they can induce DDIs [114], hepatic encephalopathy, and are contraindicated in patients with liver failure [73] (Table 5). Among this class, morphine (which undergoes glucuronidation and is affected later by hepatic insufficiency in comparison with the CYP450 system) and intravenous fentanyl appear to be the safest options. 

Finally, muscle relaxants are variously involved in liver failure (Table 5). Metaxalone is contraindicated in patients with significantly impaired hepatic disease [49]. 

In this scenario, it is not futile to remember that the concomitant consumption of possible hepatotoxic drugs such as statins, antimicrobials, amiodarone, allopurinol, and contraceptives must be taken into account [88,116].

**Table 5 pharmaceutics-15-01165-t005:** Effects of analgesic drugs in patients with hepatic failure.

Drug	Biliary Excretion	Effect in Patients with Hepatic Insufficiency
Acetaminophen	1–10% [102]	Contraindicated in patients with severe hepatic insufficiency. Caution is needed in the cases of mild and moderate hepatic insufficiency [63].
Oxycodone	Not clearly estimated but relevant; important hepatic metabolism [73,117]	Oxycodone is contraindicated in patients with moderate–severe hepatic impairment on the label. Some authors refer to the necessity to reduce dosage and prolong intervals [73].
Buprenorphine	70% [118]	Buprenorphine should be used with caution in cases of mild–moderate hepatic impairment and is contraindicated in severe forms [70].
Fentanyl	Little biliary excretion but strong hepatic metabolism [119]	Concerning fentanyl, dose reduction may be necessary [74].
Hydromorphone	1% in feces [107]	Hydromorphone is contraindicated in patients with severe hepatic impairment, whereas dose reduction is suggested in those with moderate impairment, and caution is needed in those with mild impairment. It should be avoided in patients with hepatorenal syndrome [108].
Methadone	10–45% of the metabolite [101]	Methadone is contraindicated in patients with severe hepatic impairment. Caution (lower doses and prolonged intervals between administration) is needed in those with mild–moderate illness, even if other authors describe no dose adjustments [66,104,114].
Tapentadol	1% [120]	Tapentadol is contraindicated in patients with severe hepatic impairment. No dose adjustments are required in those with mild hepatic impairment, whereas low doses and prolonged dosing intervals are recommended in patients with moderate illness [75].
Tramadol	10% [121]	Tramadol is not recommended in patients with severe hepatic impairment, and some authors suggest the prolongation of dosing intervals or dose reduction in those with mild–moderate forms [72,114]. American label suggests that the recommended dose for adult patients with severe hepatic impairment is 50 mg every 12 h [109].
Hydrocodone	Low biliary excretion, data not available; relevant hepatic metabolism [71]	Dose reduction [77].
Morphine and codeine	5–10% morphine in feces [67]; similar percentages for codeine, which is then converted into morphine [106]	Morphine is contraindicated in patients with severe hepatic impairment, and dosage should be reduced by 25% in those with moderate hepatic impairment [67]. Other authors suggest dose reduction and prolongation of dose intervals for oral formulation and dose reduction only for intravenous formulation. It should be avoided in hepatorenal syndrome [114]. Codeine use is not recommended for the possible lack of analgesic effect [114].
Duloxetine	20% [110]	Duloxetine must not be used in patients with hepatopathy and alterations in hepatic function. Moreover, this drug is hepatotoxic [41,114].
Amitriptyline	Small quantity [31]	Amitriptyline is contraindicated in liver diseases [31]. Other authors suggest its use with caution, even if it is worse-tolerated than nortriptyline and desipramine [114].
Lidocaine 5% patch	Minor quote	Severe hepatic impairment: caution [112].
Tizanidine	20% [122]	Tizanidine is generally contraindicated in patients with relevant hepatic compromise. It should be used only if the benefit outweighs the risk [46].
Baclofen	25% [123]	Baclofen is not metabolized by liver, but it is hepatotoxic: caution is needed [44].
Thiocolchicoside	80% [48]	It may increase liver enzymes or cause hepatic damage [48].
Cyclobenzaprine	Minor quote	It may increase liver enzymes or cause hepatic damage [45].
Eperisone	24.4% [47]	Eperisone is contraindicated in patients with severe hepatic failure, and caution/dose adjustment may be needed in other forms (maximum 150 mg daily dose) [47].
Pregabalin	None	No dose adjustment [19,30].
Gabapentin	None	No dose adjustment [19,30].

### 4.3. Hypertension

Managing patients with pain and hypertension is common. Some drugs including duloxetine (which is contraindicated in patients with uncontrolled blood pressure levels), muscle relaxants, gabapentin, NSAIDs, tramadol, and tapentadol may increase blood pressure levels [19,42,72,75,124]. Other substances such as other opioids, baclofen, cyclobenzaprine, and tizanidine may decrease blood pressure [44,46,73]. Therefore, it is important to acquire basal blood pressure levels and monitor them throughout the treatment duration.

### 4.4. Bone Fracture 

Opioids and NSAIDs can increase the risk of bone fragility [90]. A meta-analysis by Ping et al. [125] showed an increased rate of hip fracture in opioid users. A retrospective study by George and colleagues evidenced an increased risk of non-union with COX-2 inhibitors (or other NSAIDs acting on COX-2 activity) and opioids but not with other NSAIDs [126]. The coadministration with other drugs associated with fracture risk (e.g., glucocorticoids, proton pump inhibitors, loop diuretics, nitrates, SSRI/SNRIs, or sedatives) may facilitate this eventuality [127]. Opioids may generate bone metabolism alteration through their deep effects on the endocrine system [125]. Although drugs such as proton pump inhibitors may increase fracture risk through a biochemical mechanism, opioids, other CNS-acting medications, and antihypertensive drugs (also acting on calcium excretion) determine falls associated with traumas and injuries [127].

### 4.5. Cardiovascular Toxicity

Heart rate and QT interval may be affected by different drugs implicated in pain control, and patients suffering from arrhythmias may suffer due to this. Amitriptyline, cyclobenzaprine, and eperisone are associated with tachycardia or palpitations [31], whereas opioids can induce both bradycardia [128] and tachycardia [72,73]. Amitriptyline and cyclobenzaprine are contraindicated in patients with cardiac insufficiency, rhythm alterations, and in the post-ischemic period [31]. Cyclobenzaprine is also contraindicated in those with hyperthyroidism, myocardial infarction, monoamine oxidase inhibitors coadministration, and arrhythmias [45]. 

### 4.6. Vertigo

Patients with pain and vertigo are not easy to treat since the use of some pain medications may increase the possibility of falls. All the CNS-acting drugs including anticonvulsants (in particular, pregabalin and gabapentin), antidepressants, opioids, and muscle relaxants share this risk, and the management of these patients may not be easy [31,41,49,90]. The resolution of the underlying vestibular pathology and the eventual choice of therapeutic options to manage vertigo must take into account possible interactions with pain medications [55]. It is interesting to observe that amitriptyline acts as an antagonist on histamine receptors, increasing somnolence [31], whereas some opioids (mainly morphine and codeine) may increase histamine release [76]. Antihistamines are part of the management of dizziness and, particularly for amitriptyline, coadministration may be a matter of interaction [55]. 

### 4.7. Respiratory Diseases

Patients with chronic obstructive pulmonary disease, asthma, and obstructive sleep apnea syndrome should avoid opioids [73,129]. The concomitant administration of CNS-depressing drugs may be dangerous in reducing the respiratory drive [130,131].

### 4.8. Gastrointestinal Diseases

NSAIDs are associated with gastrointestinal bleeding and ulcers. COX-2 inhibitors seem to be the safest option. However, they are associated with minor but significant gastrointestinal toxicity according to COX-2’s role in mucosal repair [132]. Amitriptyline, duloxetine, muscle relaxants, pregabalin, and gabapentin (this last drug may also produce gingivitis) are commonly associated with mild to moderate gastrointestinal symptoms including diarrhea, stypsis, dyspepsia, vomiting, and nausea [19,30,31,41,44,45,46,48]. Amitriptyline is associated with stypsis, according to its anticholinergic effect [31], and may interact with opioids. Antidepressant drugs are contraindicated in patients with pyloric stenosis and other similarly obstructive conditions [31]. Stypsis is a crucial issue in this setting since opioids are commonly associated with this side effect [133]. Using the lowest opioid dosage and managing stypsis through dietary changes, stool softeners, or, eventually, laxatives may be a proper strategy. It is important to note that tapentadol is associated with a reduced rate of constipation in comparison with other opioids. Furthermore, long-acting formulations are more probably related to this adverse event compared with short-acting ones. Opioid rotation may minimize this adverse event. The use of peripheral μ-opioid receptor antagonists (PAMORAs) such as naloxegol, naltrexone, and naldemedine is another useful option [49,134,135]. Oxycodone is a particular drug since it is a weak/moderate opioid at the dosage of 5 mg but a strong opioid at higher doses. Oxycodone was associated with a rate of constipation of 6.1% in a retrospective study by Staats and colleagues [136]. Higher dosages are generally associated with an increased probability of adverse events. Interestingly, the co-formulation of oxycodone and naloxone (a μ-opioid receptor antagonist) is a clinically relevant option to minimize gastrointestinal side effects (stypsis) and increase therapeutic adherence [137]. 

### 4.9. Sexual Dysfunctions 

Pain may impair the psychosocial life of an individual, also affecting his relationships [3]. Long-term use of opioids can increase the release of prolactin (PRL) and reduce the release of GnRH (gonadotropin-releasing hormone) with the development of gynecomastia, erectile dysfunction, and reduction in sexual desire in men and amenorrhea in women. Therefore, hypogonadism and endocrine alterations are crucial issues, especially in chronic high-dose consumption [73,138]. Furthermore, amitriptyline, duloxetine, gabapentin, and pregabalin are commonly related to erectile dysfunction. These side effects have rarely been described during muscle relaxants treatment [44,45]. Other adverse events such as amenorrhea and gynecomastia are less frequent but possible with the aforementioned substances [19,30,31,41]. Patients with erectile dysfunction (e.g., with diabetic neuropathy-vasculopathy) consuming other sexuality-affecting medications (e.g., β-blockers) may require deprescription or a change in treatment. For example, nebivolol is the more indicated β-blocker in erectile dysfunction patients due to its action on nitric oxide [139]. The complaint of gynecomastia with drugs such as spironolactone, calcium antagonists, some antibiotics, efavirenz, or antipsychotics is a difficult issue [140]. Similar sexuality concerns are possible with women affected by menstrual cycle alterations [138].

### 4.10. Urinary Symptoms 

Patients with benign prostatic hyperplasia (BPH) or urinary obstructive diseases may experience a worsening of their symptoms with amitriptyline, eperisone, or cyclobenzaprine [31,45,47]. Duloxetine and baclofen (also determining enuresis) are frequently associated with dysuria and pollakiuria [41,44]. Opioids have minimal anticholinergics effects, and their binding to spinal receptors may determine bladder relaxation effects and a low rate of urinary retention [141]. 

### 4.11. Other Clinical Conditions

Patients affected by glaucoma must avoid amitriptyline, whereas caution is needed with duloxetine and cyclobenzaprine [31,41,45]. Amitriptyline and baclofen are (rarely) sometimes associated with variations of glycemia [31,44]. 

## 5. Discussion

In the management of nociceptive or neuropathic pain, opioid use can induce the development of severe adverse events requiring careful monitoring [90]. 

Gabapentin and pregabalin, used for neuropathic and nociplastic pain, have similar indications and side effects (e.g., dizziness, diplopia, blurred vision, and psychiatric, neurological, and cutaneous adverse events). However, gabapentin is related to more side effects (infections, leukopenia, anorexia, and increased appetite) and to a more complex administration scheme [19,20]. Furthermore, these drugs seem to be ideal in poly-treated patients but have an important potential for psychiatric adverse events [142]. Pregabalin seems more effective for central neuropathic pain and fibromyalgia [143]. According to a meta-analysis by Salerno et al. [144], antidepressants seem to be more effective for relieving pain symptoms in patients with neuropathic pain with a worse functional outcome.

The concomitant administration of more than one pain drug is useful in order to reduce dosage and side effects, but clinicians must be aware of eventual interactions [12,49]. 

The role of transporters in DDI is complex. For example, fentanyl, methadone, and morphine are substrates of P-glycoprotein, and this may increase the risk of interactions. The presence of multiple pharmacogenomic polymorphisms may strongly influence drug response and therapeutic efficacy [145,146].

Concerning nociplastic or primary chronic pain, they are often difficult to treat. A review of non-opioid pharmacological agents for the management of various chronic pain conditions (including fibromyalgia, low back pain, and chronic headaches) reported mostly small improvements (e.g., 5–20 points on a 0–100 scale) for gabapentinoids, SNRIs, and NSAIDs for pain and function in the short term, with intermediate and long-term outcomes infrequently assessed [29]. In the SIGN guidelines, fluoxetine (and other antidepressants such as citalopram, escitalopram, and sertraline in the NICE guidelines) may be considered for chronic primary pain/fibromyalgia [12].

Opioids can be used in the management of nociceptive and neuropathic pain, but their use is related to the development of tolerance and dependence and is also contraindicated in several clinical conditions (severe respiratory insufficiency, acute alcoholism and delirium tremens, CNS drugs, children and adolescents of <16 years, breastfeeding, severe CNS compromise, paralytic ileus, acute abdominis, pulmonary heart disease, and chronic stypsis) [70]. Codeine is also contraindicated in CYP2D6 ultrarapid metabolizers [65]. Tramadol is contraindicated in epilepsy and MAOI (monoaminoxidase inhibitors) concomitant treatment [72]. Hydromorphone is also contraindicated in severe gastrointestinal stenosis, MAOI treatment, and coma [108]; methadone in cardiopathy, uncontrolled diabetes, porphyria, hypotension, hypovolemia, intracranial hypertension, or cranial traumas [66]; and morphine in intracranial hypertension, cranial traumas, uncontrolled epilepsy and convulsion, MAOI treatment, and biliary surgery [67]. 

Several other compounds can be used in the management of pain even if their use is not reported in guidelines.

Cannabis-based medicines hold promise but require formal studies before their widespread use in patients at high risk for adverse effects. Products are labeled on the basis of tetrahydrocannabinol and cannabidiol contents; these compounds are indicated for the management of neurodegenerative muscle diseases but not for pain even if efficacy has been suggested in patients with nociplastic and neuropathic pain [147]. A systematic review and meta-analysis by Aviram and Samuelly-Leichtag showed that cannabis may have a certain efficacy in patients with chronic pain, especially neuropathic and nociplastic pain, and that inhalation rather than oral consumption may reduce gastrointestinal adverse events [148]. Adverse effects include an increase in heart rate (and risk of myocardial infarction), dizziness, seizure, psychosis, dependency, euphoria, or other psychiatric adverse events [49].

Palmitoylethanolamide (PEA) and acetyl-L-carnitine (ALC) may be used in patients with nociplastic and neuropathic pain [149]. L-acetyl-carnitine is an endogenous substance that modulates pain via different mechanisms: stimulation of mitochondrial function and repair factors such as nerve growth factor, antioxidant activity, and activation of metabotropic glutamate receptor 2 (mGlu2). An experimental study by Cuccurazzu et al. [150] showed that ALC activates NF-kB, increasing the expression of mGlu2 and suggesting a pro-neurogenic effect. Experimental data show that ALC modulates neuropathic pain, and multiple administration is necessary to obtain analgesia [151]. Similar results were obtained by Parisi et al. [152], evidencing an excellent safety profile and good efficacy. However, Rolim and colleagues assessed the uncertainty of its efficacy in patients with diabetic neuropathy [153]. Gastrointestinal side effects are the most commonly reported side effects [154]. PEA exerts its action on the endocannabinoid system and reduces inflammation. This molecule stimulates the effects of endo- or phytocannabinoids acting on peroxisome proliferator-activated receptor α (PPAR-α), transient receptor potential vanilloid type 1 (TRPV1), and cannabinoid receptors. Moreover, it reduces the activity of inflammatory enzymes and mast cell degranulation. 

Emerging treatments such as neuromodulation have theoretical potential but have little data on their effectiveness. This therapy includes deep brain and motor cortex stimulation, non-invasive treatments (transcranial magnetic stimulation, transcranial direct-current stimulation, and transcutaneous electrical nerve stimulation), and peripheral nerve stimulation. Low–moderate quality evidence is available for peripheral nerve stimulation for neuropathic pain. However, few trials have been conducted, and some of these techniques are not approved for clinical use [155].

Other procedural (exercise programs, psychological therapy, and social interventions), physical (transcutaneous electrical nerve stimulation (TENS), ultrasound, interferential therapy, manual therapy, bracing, cold, and heat), or interventional treatments (radiofrequency, acupuncture, and corticosteroid–anesthetics injection at various levels including epidural steroid injections, sacroiliac joint corticosteroid–anesthetics coadministration, botulinum injections, cryoneuroablation, thermal intradiscal procedures, peripheral nerve blocks, sympathetic nerve blocks, intrathecal medication pumps, joint injections, and vertebral augmentation) are other therapeutic opportunities depending on the clinical setting, guideline recommendations, and data from larger clinical trials [49,82,92].

Other techniques are arising. Diamagnetic therapy using pulsed magnetic fields may display an important activity on nociceptive, neuropathic, and nociplastic pain through its effect on inflammatory cytokines, neuromodulation, and neuroprotective effects. Moreover, its effect on liquids is very useful for edematous conditions [156,157,158,159]. We observed that the use of a high-intensity low-frequency magnetic field was effective in patients with ulcers and complex regional syndrome, reducing pain levels and improving their clinical status [160,161].

Oxygen–ozone therapy may be useful in the management of patients with osteoarthritis and low back pain, whereas other musculoskeletal disorders must be studied better. The mild oxidative stress induced by oxygen–ozone may activate the Nrf2 protein, through its separation from Kelch-like ECH-associated protein 1 (Keap-1). This effect leads to an increase in transcription mediated by Nrf2, which enhances genes involved in inflammation reduction and oxidative stress response. Moreover, oxygen–ozone may have an important effect on hypoxic tissues via its action on prostaglandins, nitric oxide, and adenosine, which leads to vasodilation. Lastly, the increased production of 2,3 diphosphoglycerate and lipid peroxidation may shift the hemoglobin dissociation curve to the right [162]. 

Finally, an important point in the management of chronic pain is represented by the biopsychosocial approach. In this model, pain is viewed as a dynamic interaction among and within the biological, psychological, and social factors unique to each person. According to this model, the treatment of the “whole” person (considering anxiety, depression, and stress) is far more important than focusing merely on a disease. This approach could be an add-on treatment to common drug use. A Cochrane review reported that cognitive behavioral therapy has small beneficial effects in reducing pain, disability, and distress, whereas the benefits from behavioral therapy were uncertain because of the poor quality of the studies included [163].

Our review has some limitations. Firstly, we did not include some non-pharmacological treatments such as acupuncture and bidimensional techniques since they were not included in the guidelines. The beginning of more solid studies showing the relevance of their biological effects is warranted. Secondly, many of the cited guidelines did not indicate the pain level/severity for which pharmacological treatment is recommended. This fact allows subjective decisions that could be a limit in clinical practice. Moreover, the synergic role of drugs and the importance of gender difference is not clarified in the clinical guidelines. In this context, the role of the volume of distribution has not been evaluated. These considerations may be key points in future recommendations to reduce drug dosage and obtain appropriate therapy. The guidelines did not establish an adequate or ideal dosage for each patient but simply describe an optimal range, including the initial dosage and maximum posology. Moreover, we did not enclose in this review chronic pain in patients with cancer or headache, and these are very important points that need a more appropriate evaluation. Finally, we did not evaluate the role of gender (e.g., economic, political, and religious status, and sex) in chronic pain as well as the role of genetic factors. To date, the guidelines do not consider this aspect; as a result, to date, it is very hard to obtain personalized therapy.

In conclusion, even if the guidelines represent an important guide for the common treatment of patients with pain, they have several limitations: (i) they do not evaluate the characteristics of each patient (age, gender, comorbidity, and polytherapy); (ii) there is no comparison with a single intervention at the time of their publications; (iii) they do not evaluate the possibility of drug interactions and genetic variability; (iv) very few guidelines evaluate nociplastic pain, and no guidelines evaluate pain combinations; and (v) no guidelines report the role of alternative treatments (other compounds or techniques). Therefore, the guidelines must be revised considering these factors and the possibility of integrating these suggestions into daily clinical activity even if there is no strong evidence for the absence of double-blind randomized clinical trials. A crucial point is the pathophysiological and diagnostic comprehension of pain and understanding the environmental, genetic, and pathological reasons that cause symptomatology. This allows physicians to make the right decisions, taking into account patients’ genders, comorbidities, and concomitant medications.

## Figures and Tables

**Figure 1 pharmaceutics-15-01165-f001:**
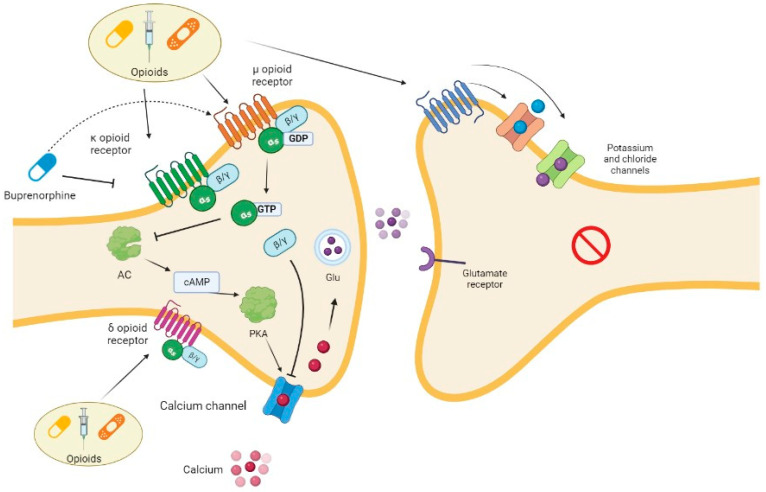
Opioid mechanism of action. Opioids bind to their μ, κ, and δ receptors at presynaptic level, carrying out different actions. After the interaction with a receptor, the α subunit of protein G inhibits the pathway of AC, resulting in a reduction in calcium channel activity and then the release of glutamate. The same channel is inhibited via the βγ subunit. Buprenorphine is a particular drug since it has partial agonist activity on μ receptor and antagonist activity on κ receptors. Opioids also exert stimulating activity on calcium and chloride channels, resulting in hyperpolarization at postsynaptic level. AC, adenylate cyclase; cAMP, cyclic adenosine monophosphate; GDP, guanosine diphosphate; Glu, glutamate; GTP, Guanosine-5′-triphosphate; PKA, protein kinase A.

**Figure 2 pharmaceutics-15-01165-f002:**
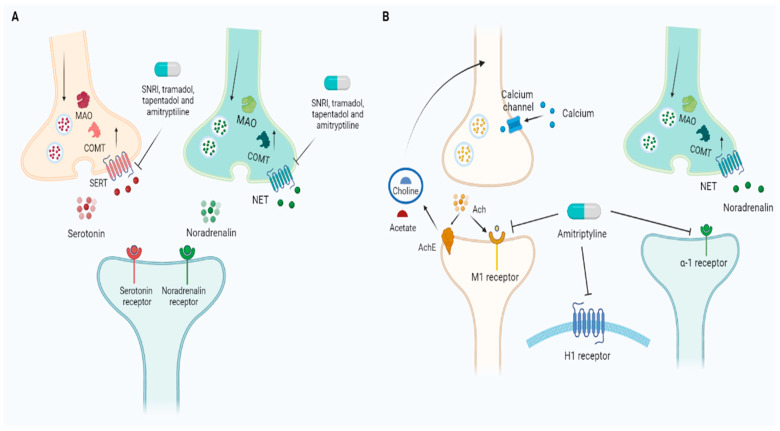
Antidepressants used for pain management. (**A**) Amitriptyline and SNRIs inhibit SERT and NET, blocking serotonin and noradrenaline reuptake and increasing the availability of the two neurotransmitters in synaptic cleft. The same action is shared by two opioids, non-antidepressant drugs, namely, tramadol and tapentadol. (**B**) Nevertheless, amitriptyline is associated with several side effects according to its inhibitory action on cholinergic, adrenergic, and histaminergic pathways. Ach, acetylcholine; AchE, acetylcholinesterase; COMT, catechol-O-methyltransferase; MAO, monoamine oxidase; NET, norepinephrine transporter; SERT, serotonin transporter; SNRI, serotonin and norepinephrine reuptake inhibitors.

**Figure 3 pharmaceutics-15-01165-f003:**
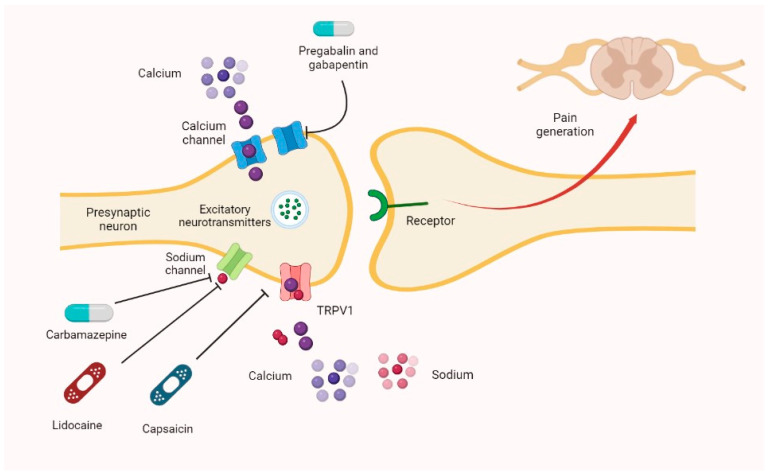
Other neuropathic pain drugs’ mechanisms of action. The main principle of counteracting neuropathic pain is reducing the release of excitatory neurotransmitters in the synaptic cleft. Carbamazepine and lidocaine inhibit sodium channels, whereas capsaicin exerts its activity on TRPV1. Pregabalin and gabapentin block calcium channels in their α2δ subunit. TRPV1—transient receptor potential cation channel subfamily V member 1.

**Table 1 pharmaceutics-15-01165-t001:** Opioids characteristics [11,12]. SNRIs: serotonin–norepinephrine reuptake inhibitors.

Characteristic	Opioids
Weak	Codeine, tramadol, hydrocodone, and dihydrocodeine
Strong	Morphine, oxycodone, fentanyl, buprenorphine, hydromorphone, methadone, and tapentadol
Antagonist	Naloxone and naltrexone
Nociceptive pain	High activity: codeine, methadone, hydrocodone, hydromorphone, and morphine Low activity: tramadol, oxycodone, and fentanyl
Neuropathic pain	Tramadol, oxycodone, fentanyl, buprenorphine, and tapentadol
CYP2D6 metabolism	Codeine, tramadol, oxycodone, and hydrocodone
CYP3A4 metabolism	Buprenorphine, hydrocodone, methadone, oxycodone, tramadol, and fentanyl
Liver conjugation	Buprenorphine, codeine, hydromorphone, morphine, oxycodone, tapentadol, and tramadol
Mechanism of action	μ receptor full agonist (morphine, oxycodone, fentanyl, hydromorphone, hydrocodone, methadone, tapentadol, and tramadol) k receptor agonist (oxycodone) μ receptor partial agonist and k receptor antagonist (buprenorphine) SNRI activity (tapentadol and tramadol)
Kidney excretion	Buprenorphine (30%), codeine (90%), fentanyl (75%), hydrocodone (6.5% of the parental drug, higher quote including metabolites), hydromorphone (90%), methadone (30%), morphine (90%), oxycodone (80%), tapentadol (99%), and tramadol (90%)
Liver excretion	Buprenorphine (70%), codeine (10%), fentanyl (9%), hydrocodone (data not available), hydromorphone (62% of oral dose eliminated by first-pass; 1% in feces), methadone (50%), morphine (10%), oxycodone (20%), and tramadol (10%)

**Table 2 pharmaceutics-15-01165-t002:** Possible drug interactions in patients with chronic pain using analgesic or anti-inflammatory drugs.

Drugs Used for Pain Management	Interacting Drug	Comment
Anticonvulsants (pregabalin and gabapentin)	CNS-depressing drugs (e.g., opioids) [19,30] and alcohol	Respiratory depression risk. If possible, avoid concomitant use, or reduce dosage.
Gabapentin	Antiacids containing aluminum and magnesium [19,30]	Reduction in gabapentin bioavailability.
Amitriptyline	CYP2D6, CYP2C19, CYP3A4, and CYP1A2 inhibitors and inducers [31]	Relevant variations in bioavailability are possible.
	Ethanol [32]	Increase in amitriptyline concentration.
	QT-increasing drugs	Risk for arrhythmias [31].
	Valproic acid	Increase in amitriptyline concentration.
	Antihypertensive drugs	Risk of further decrease in blood pressure due to α-1 receptor antagonism, but cases of hypertension have been described [31,33].
	Anticholinergic drugs	Increase in the side effects related to anticholinergic actions of amitriptyline [31,33].
	CNS-depressing drugs and alcohol	Increase in CNS depression [31].
	L-DOPA and phenylbutazone	Reduced gastric emptying may arise: L-DOPA and phenylbutazone may be inactivated for this reason. Furthermore, L-DOPA coadministration may facilitate arrhythmias and hypotension [31]. Nevertheless, L-DOPA and amitriptyline coadministration has been associated with better molecular efficacy in Parkinson’s disease [34].
	Antihistamines	Possible increase in QT interval and increased sedation [35,36,37].
Duloxetine	Other antidepressants or drugs increasing serotonin levels (e.g., tramadol and tapentadol)	Serotonin syndrome risk [38,39].
	CYP1A2 and CYP2D6 inhibitors or inducers	Possible variation in duloxetine levels [40]. Contraindicated if CYP1A2 inhibitors are being used in therapy [41].
	CYP2D6 substrates	Increase in these drugs’ levels due to the moderate inhibitory action of duloxetine on CYP2D6 [40].
	CNS-depressing drugs and alcohol	Increase in CNS depression risk [40].
	Antihypertensive drugs	Increase in blood pressure due to the action on noradrenalin reuptake [42].
	Anticoagulants or antiaggregant drugs	Increase in bleeding risk related to action on platelet serotonin [40,43].
Muscle relaxants	Antihypertensive drugs	Various interactions are described since baclofen, tizanidine, and cyclobenzaprine may decrease blood pressure. No alterations are described with eperisone and thiocolchicoside [44,45,46,47,48].
	CNS-depressing drugs and alcohol	Increased CNS depression [49].
Baclofen	Tricyclic antidepressants	Possible increase in muscular hypotonia risk [44].
	Carbidopa and L-DOPA	Worse control of Parkinson’s symptoms. Confusion, hallucinations, and headache [44].
	Lithium	Increase in hyperkinetic symptoms [44].
	Drugs decreasing renal function	Increase in baclofen levels [44].
Cyclobenzaprine	Structural analog of tricyclic antidepressants [45]	Similar pharmacodynamic actions are expected, including sedation, anticholinergic effects, and blurred vision.
Eperisone	Calcium antagonists [47]	Increased calcium antagonists’ effects.
	Salicylates [47]	Reduced salicylates levels.
Metaxalone	Drugs increasing serotonin levels	Possible risk of serotoninergic syndrome [50].
Methocarbamol	Pyridostigmine	Decreased effect of pyridostigmine in patients with myasthenia [51].
Tizanidine	CYP1A2 inhibitors and inducers [46]	Increased/decreased levels of tizanidine. Contraindicated in presence of CYP1A2 inhibitors [46].
	Drugs prolonging QT [46]	Risk for QT prolongations.
	Oral contraceptives, verapamil, and cimetidine [46,49]	Possible increase in tizanidine levels.
	Beta-blockers or digoxin [46]	Possible increase in hypotension and bradycardia rate.
NSAIDs	CYP2C9 inhibitors/inducers [52,53]	Evaluate dosage increase/reduction.
	Aspirin and other associated NSAIDs	Risk for reduced effect of aspirin [54].
	Antihypertensive drugs due to kidney damage and inhibition of natriuretic response to diuretic response, impaired synthesis of prostaglandins, sodium and water retention, and suppression of plasma renin activity [55,56]	Increase in blood pressure levels. Minor interactions are described with calcium antagonists [53].
	Anticoagulants, antiaggregant drugs, corticosteroids, SSRIs, and even nutraceuticals/supplements such as Ginkgo Biloba [57,58,59]	Increase in hemorrhagic risk. Warfarin may be released from albumin after NSAID coadministration. Moreover, concomitant CYP2C9 metabolism by the two drugs may affect their concentrations [58,59].
	Lithium, methotrexate, zidovudine, and digoxin	NSAIDs may reduce kidney elimination of some substances including lithium and methotrexate [58,60].
	Probenecid	Aspirin may reduce probenecid effects [58].
	Nephrotoxic medications (e.g., tacrolimus, aminoglycosides, and ciclosporin) [61]	Increased nephrotoxicity.
	Zidovudine	Increase in NSAIDs’ plasma levels (diclofenac in particular) and toxicity in animal models [62]. In humans, naproxen modified zidovudine conversion to its glucuronidated metabolite (GZDV) with a transformation in toxic metabolites.
Acetaminophen	Chloramphenicol	Possible increase in chloramphenicol half-life [63].
	Drugs slowing or fastening gastric emptying and cholestyramine	Possible increase or reduction in bioavailability. Cholestyramine may reduce paracetamol’s absorption [63,64].
	Hepatotoxic drugs	Increased risk for transaminases increases or liver failure. Phenytoin may reduce paracetamol efficacy and increase liver failure risk [65].
Opioids	Antidiarrhoeic drugs,	Stypsis [66].
	CNS-depressing drugs, and alcohol	CNS depression [67,68].
	CYP3A4 inhibitors/inducers	Buprenorphine, hydrocodone, methadone, oxycodone, tramadol, and fentanyl may be variously involved in these reactions with increases or reductions in drug levels [66,69,70,71,72,73].
	CYP2D6 inhibitors	Codeine, tramadol, oxycodone, and hydrocodone may be involved. Codeine, a prodrug, may be counteracted in its therapeutic action [65,69,71,72,74].
Methadone	Ammonium chloride	Ammonium chloride may facilitate methadone (a weak base) elimination via its action on urine pH [66].
	Anticholinergic drugs	Increase in anticholinergic effects, stypsis in particular [66].
	Desipramine	Increase in desipramine levels [66].
	Didanosine, stavudine, and zidovudine	Methadone may reduce didanosine and stavudine bioavailability, affecting their absorptions and first pass metabolisms. Nevertheless, methadone may increase zidovudine levels, reducing glucuronidation processes and, therefore, its renal clearance [66].
	Octreotide	Possible reduction in analgesic effect.
	P-gp inhibitors and inducers	Methadone is a P-gp substrate. Therefore, the inhibition/induction of this protein may result in variations in methadone’s serum levels [66].
	Drugs prolongating QT or antiarrhythmics	Possible risk for arrhythmias [66].
Morphine	Cimetidine	Reported cases of confusion and respiratory depression [67].
	Diuretics	The increase in ADH may contrast the effect of diuretics [67].
	Muscle relaxants/blockers and oral anticoagulants	Morphine may increase the effects of these drugs [67].
Oxycodone	Anticholinergic drugs	Increase in anticholinergic effects [73].
Fentanyl, methadone, oxycodone, tapentadol, and tramadol	Drugs increasing serotonin levels	Serotonin syndrome risk. Tapentadol has minor action on serotonin reuptake; therefore, the risk of serotonin syndrome is minor, compared with tramadol [72,75]. Methadone has an elevated potential of determining this syndrome [76]. A theoretic risk is also reported for hydrocodone and buprenorphine [77,78].
Tapentadol (alone)	Naproxen and probenecid	These drugs may increase tapentadol levels but without clinical significance. Due to its glucuronidation related metabolism, tapentadol shows few interactions [75,79].
Tramadol (alone)	Warfarin and coumarin derivatives	Possible increase in INR [72,80].
	Ondansetron	Possible necessity to increase tramadol dose [72,81].

**Table 3 pharmaceutics-15-01165-t003:** International guidelines for chronic pain. SIGN: Scottish Intercollegiate Guidelines Network; CDW: Colorado Division of Workers; AGS: American Geriatric Society; COX: cyclooxygenase; DHHS: Department of Health and Human Services; NeuPSIG: Neuropathic Pain Special Interest Group; NICE: National Institute for Health and Care Excellence; NMDA: N-methyl-D-aspartate; NSAIDs: non-steroidal anti-inflammatory drugs; OTC: over the counter; SNRI: noradrenaline–serotonin reuptake inhibitor; TCA: tricyclic antidepressant.

	STEP I	STEP II	STEP III	STEP IV	STEP V	STEP VI
**Nociceptive pain**						
SIGN [12]	Paracetamol or NSAIDs	Weak opioids or topical NSAIDs	Strong opioids			
CDW [49]	NSAIDs or COX-2 inhibitors					
AGS [91]	Paracetamol (up to 4 g/day)	NSAIDs	Opioids			
DHHS [92]	Ia: paracetamol Ib: ibuprofen or naproxen Ic: paracetamol plus ibuprofen or naproxen	IIa: codeine IIb: tramadol	IIIa: low-dose morphine or buprenorphine patch (if morphine is ineffective) IIIb: high-dose morphine or 5–30 mg of oxycodone twice a day or fentanyl/buprenorphine patch if morphine is ineffective IIIc: 50 mg of tapentadol twice daily			
ESCEO [93]	Chondroitin sulfate or glucosamine sulfate	Paracetamol or topical NSAIDs	NSAIDs	Intra-articular injection of hyaluronic acid or corticosteroids	Duloxetine	Surgery
**Neuropathic pain**						
SIGN [12]	Amitriptyline or gabapentin	Pregabalin	SNRIs	5% lidocaine	Opioids	9% capsaicin
CDW [49]	Tricyclic antidepressants	Gabapentin/pregabalin or SNRIs (duloxetine)	Other anticonvulsants	Low-dose opioids		
AGS [91]	Duloxetine or pregabalin					
DHHS [92]	Amitriptyline/imipramine	Gabapentin (1st line) or pregabalin (2nd line) or 0.075% capsaicin cream	Duloxetine or lidocaine plasters (5%-700 mg/plaster) or capsaicin patch (8%-179 mg/plaster)			
Practice [94]	Duloxetine or TCA	Lidocaine or ketamine				
NICE [2].	Antidepressants	Gabapentin or pregabalin				
NeuPSIG [14]	TCA, SNRI, or gabapentin/pregabalin	Tramadol, lidocaine, and capsaicin patches	Opioids or botulin toxin-A			

**Table 4 pharmaceutics-15-01165-t004:** Effects of analgesic drugs in patients with renal failure. eGFR: glomerular filtrate rate.

Drug	Kidney Excretion	Effects in Patients with Renal Failure
Acetaminophen	90–99% [102]	Not used with eGFR of <10 mL/min [63,100].
Oxycodone	50%	Dose adjustments [73]. Some authors consider oxycodone unsafe in patients with advanced kidney failure due to its accumulation risk, interactions, and CYP450 polymorphisms [100].
Buprenorphine	10–30 [103]	Caution with eGFR of <30 mL/min [70].
Fentanyl	10% or less of active compound and 75% of the total dose. Metabolites are excreted mainly in urine [74].	Dose monitoring [74].
Methadone	20–50% as methadone or its metabolites [101]	Contraindicated in patients with severe kidney impairment [66]. Lower doses and longer intervals between administration in patients with kidney impairment [104].
Morphine	70–80% [105]	eGFR of 10–50 mL/min: dose reduction of 25%; eGFR of <10 mL/min: dose reduction of 50% [104]. One of the worst options in advanced kidney failure due to accumulation risk [100].
Codeine	Mainly excreted in kidneys [106]	Caution is needed. Davison et al. consider codeine one of the worst options in patients with advanced kidney failure due to CYP2D6 polymorphisms and accumulation risk [100].
Hydromorphone	Most of the dose; 7% unmodified drug [107]	Dose reduction [108].
Hydrocodone	Eliminated with its metabolites, mainly in kidneys, percentage not available [71]	Caution/dose reduction [71,77]. Davison considers it one of the worst options in patients with advanced kidney failure, according to CYP2D6 polymorphism-related and variable responses and possible accumulation risk [100].
Tapentadol	99% [75]	Not recommended in patients with severe insufficiency [75].
Tramadol	90% [72]	Prolonged interval between doses; do not use long-release formulation [72]. Increase the interval of administration to 12 h, and limit maximum daily dose to 200 mg [109].
Duloxetine	70% [110]	eGFR of <30 mL/min: do not use [41].
Amitriptyline	95% [31]	No dose reduction [31,100,111].
5% lidocaine patch	>85%	eGFR of <30 mL/min (severe kidney impairment): caution [112].
Tizanidine	60–70% [46]	eGFR of <25 mL/min: start with 2 mg/day [46].
Baclofen	75% [44]	Start with lower dosages in all patients with mild–moderate kidney impairment, and use only if benefit outweighs the risk in those with severe kidney impairment [44].
Thiocolchicoside	20% [48]	No dose adjustments [48].
Cyclobenzaprine	80%	Low dosage [45].
Eperisone	76.6% [47]	eGFR of <25 mL/min: low dosage, max. 150 mg daily [47].
Pregabalin	99%	eGFR of 30–59 mL/min: 300 mg/daily. eGFR of 15–29 mL/min: 150 mg/daily. eGFR of <15 mL/min: 75 mg/daily [113].
Gabapentin	99%	eGFR of 30–59 mL/min: 1400 mg/daily. eGFR of 15–29 mL/min: 700 mg/daily. eGFR of <15 mL/min: 300 mg/daily [113].

## Data Availability

Not applicable.

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
