# Peer review of "The Pharmacological Treatment of Chronic Pain: From Guidelines to Daily Clinical Practice"

_pharmaceutics, 2023, doi:10.3390/pharmaceutics15041165_

Round 1

Reviewer 1 Report

The format of  tables in this paper is not well organized, I hope to improve the layout to improve the visualization.

Author Response

Reviewer 1

Dear Reviewer thank you for your comments and for suggestions that we considered in the revision of the present case report. We performed all changes in agreement with your suggestions.

Q: The format of  tables in this paper is not well organized, I hope to improve the layout to improve the visualization.

A: Table 1 (page 2), Table 5 (Page 19), Table 6 (Page 25-32) have been removed (Table 1 page 2).

 Table 3 (page 8-12), Table 4 (Page 14-18), Table 7 (Page 30-31) Table 8 (Page 32-37) have been revised

Reviewer 2 Report

Thank you for the opportunity to review this manuscript. It represents a needed review of medication options for the treatment of chronic pain. The manuscript is well written overall; however, there are a few suggestions to aid in the overall readability and usability of the information. 

* The introduction needs more background as to why this paper is needed. For example, on page 5 the authors noted differing recommendations based on different agencies. The introduction could benefit from a way to structure the manuscript. Perhaps the introduction could highlight the misleading/mixed guidelines for medication management, suggest that a critical review is necessary, and the remainder of the paper could be organized by medication class OR type of chronic pain? As of now, it feels as though information is very scattered throughout.

* Headers may be helpful once the paper has a more structured organizational flow.

* The biopsychosocial model of pain is not mentioned and would be very information to include, even in the discussion, as research on medications for pain continue to highlight only minimal effectiveness without other interventional/treatment modalities. 

* It seems as though Section 3 should be entitled common considerations for comorbid conditions. 

* There seems to be many tables, which leads to some confusion. Perhaps with more organization of the paper, the tables will have more of a theme leading to less, more structured tables.

* The conclusion is largely a repeat of some of the same concepts from the earlier text. It would be helpful to include some more critical thinking and next steps the authors suggest after reviewing the medication research.

Author Response

Reviewer 2

Dear Reviewer thank you for your comments that we considered in the revision of this manuscript. We changed it in agreement with your suggestions and we also send you a point by point rebuttal letter.

* The introduction needs more background as to why this paper is needed. For example, on page 5 the authors noted differing recommendations based on different agencies. The introduction could benefit from a way to structure the manuscript. Perhaps the introduction could highlight the misleading/mixed guidelines for medication management, suggest that a critical review is necessary, and the remainder of the paper could be organized by medication class OR type of chronic pain? As of now, it feels as though information is very scattered throughout.

Answers: Introduction has been revised in agreement with your suggestions. The remainder of the manuscript has been organized considering the type of chronic pain. In our opinion, in clinical practice the first approach is the type of pain and then we choose the drug(S).  

* Headers may be helpful once the paper has a more structured organizational flow.

Answers: all manuscript has been revised considering your suggestion

* The biopsychosocial model of pain is not mentioned and would be very information to include, even in the discussion, as research on medications for pain continue to highlight only minimal effectiveness without other interventional/treatment modalities. 

Answers: we agree with your suggestion, we added this point in the discussion (Page 44 line 1040-1048).

* It seems as though Section 3 should be entitled common considerations for comorbid conditions. 

Answers: The tile of Section 3 (page 25 line 594) has been changed considering your comment

* There seems to be many tables, which leads to some confusion. Perhaps with more organization of the paper, the tables will have more of a theme leading to less, more structured tables.

Answers: in agreement with your suggestion; Table 3 (page 8-12), Table 4 (Page 14-18), Table 7 (Page 30-31) Table 8 (Page 32-37) have been revised

* The conclusion is largely a repeat of some of the same concepts from the earlier text. It would be helpful to include some more critical thinking and next steps the authors suggest after reviewing the medication research.

Answers: it is correct, conclusion has been clarified (Page 44 and 45 line 1065-1078)

Reviewer 3 Report

The manuscript described the overall information in general, and the initial and middle parts of the manuscript should contain less information. The clinical use section needs expansion.

Some Tables do not contain novel information (Tables 1-3). Table 3 contains some incomplete information about CYPs, as these drugs are metabolised by other CYPs as well.

Figure 1 is very general, and it should be removed. Importantly Figures 1 and 2 do not fall within the scope of the manuscript as the authors should focus on chronic pain guidelines and clinical practice. 

The abstract is not organised. It must be more structured and informative. 

The manuscript does not have any discussion/opinion part. It must be included. 

The conclusion is not a conclusion, it is more like some additional information. Please revise and make it concise. 

Authors should describe the limitations of their review broadly. The review is not novel, and there are many similar manuscripts. Recently published literature should be cited in the field as well. 

Author Response

Reviewer 3

Dear Reviewer thank you for your comments that we considered in the revision of the present case report. We changed it in agreement with your suggestions and we also send you a point by point rebuttal letter.

  • The manuscript described the overall information in general, and the initial and middle parts of the manuscript should contain less information. The clinical use section needs expansion.

Answers: clinical use has been revised and clarified

Some Tables do not contain novel information (Tables 1-3). Table 3 contains some incomplete information about CYPs, as these drugs are metabolised by other CYPs as well.

Answers: Tables have been revised also considering the other referee’s comments. In particular:

Table 1 (page 2), Table 5 (Page 19), Table 6 (Page 25-32) have been removed (Table 1 page 2).

 Table 3 (page 8-12), Table 4 (Page 14-18), Table 7 (Page 30-31) Table 8 (Page 32-37) have been revised

Figure 1 is very general, and it should be removed. Importantly Figures 1 and 2 do not fall within the scope of the manuscript as the authors should focus on chronic pain guidelines and clinical practice. 

Answers: we agree with your comments and figure 1 and 2 have been removed

The abstract is not organised. It must be more structured and informative. 

Answers: abstract has been revised (page 1)

The manuscript does not have any discussion/opinion part. It must be included.

Answers: it is correct, thank you, the discussion has been revised (Page 42 and 43 line 943-997)

The conclusion is not a conclusion, it is more like some additional information. Please revise and make it concise. 

Answers: it is correct, thank you, the conclusion has been changed also considering the other referee’s comments (Page 44 and 45 line 1065-1078)

Authors should describe the limitations of their review broadly. The review is not novel, and there are many similar manuscripts. Recently published literature should be cited in the field as well. 

Answers: the limitations have been added (Page 44 line 1049-1064)